# Morphological and Immunocytochemical Characterization of Tumor Spheroids in Ascites from High-Grade Serous Carcinoma

**DOI:** 10.3390/cells12192390

**Published:** 2023-09-30

**Authors:** Simona Miceska, Erik Škof, Gorana Gašljević, Veronika Kloboves-Prevodnik

**Affiliations:** 1Department of Cytopathology, Institute of Oncology Ljubljana, Zaloška Cesta 2, 1000 Ljubljana, Slovenia; smiceska@onko-i.si; 2Faculty of Medicine, University of Ljubljana, Korytkova Ulica 2, 1000 Ljubljana, Slovenia; 3Department of Medical Oncology, Institute of Oncology Ljubljana, Zaloška Cesta 2, 1000 Ljubljana, Slovenia; 4Department of Pathology, Institute of Oncology Ljubljana, Zaloška Cesta 2, 1000 Ljubljana, Slovenia; 5Faculty of Medicine, University of Maribor, Taborska Ulica 8, 2000 Maribor, Slovenia

**Keywords:** ascites, cell blocks, high-grade serous carcinoma, ovarian carcinoma, spheroids, spheroid-associated immune cells, PD-1, PD-L1

## Abstract

Tumor spheroids in the ascites of high-grade serous carcinoma (HGSC) are poorly described. Our objective was to describe their morphological features, cellular composition, PD-1 and PD-L1 expression, and survival correlation of these parameters. The density and size of spheroids were assessed in Giemsa-stained smears; the cell composition of spheroids, including tumor cells, immune cells, capillaries, and myofibroblasts, as well as PD-1 and PD-L1 expression on tumor and immune cells was assessed in immunocytochemically stained cell block sections. Forty-seven patients with primary HGSC and malignant ascites were included. A cut-off value for a spheroid density of 10% was established, which significantly predicted overall survival. However, spheroid size did not correlate with survival outcomes. Spheroids were primarily composed of tumor cells, but the presence of lymphocytes and macrophages was also confirmed. Moreover, capillaries were present in the spheroids of three patients, but the presence of myofibroblasts was not confirmed. PD-1 was expressed on lymphocytes but not on tumor cells. PD-L1 expression was seen on both tumor and immune cells, assessed by 22C3 and SP263 antibody clones but not by the SP142 clone. Our results highlight the potential of routine cytopathological techniques to analyze spheroids in HGSC ascites as a valuable tool to investigate their potential as prognostic markers.

## 1. Introduction

Ovarian carcinoma is the most lethal gynecological malignancy in Western countries, with high-grade serous carcinoma (HGSC) as the most common and aggressive histological type, which appears to arise from the surface epithelium of the ovaries, fallopian tubes, or peritoneum [1]. Commonly, 75% of the patients are diagnosed at an advanced stage due to the lack of symptoms and reliable detection methods. HGSC is characterized by malignant ascites, which is often the first sign of the disease [2]. In the ascites, tumor cells shed from the primary tumor or visceral and parietal peritoneal carcinosis, forming free-floating spheroids [3,4]. Many studies focused on spheroids research by using 3D spheroid modeling in vitro [5,6], while our group recently showed that tumor spheroids could also be investigated from ascites itself since we confirmed they share the same immunophenotypic and molecular characteristics as the primary tumor [7]. However, currently, there are no available data regarding the quantitative and qualitative composition of spheroids from ascites and their clinical significance. To the best of our knowledge, the study of Micek et al. is the first one to systematically characterize the size of the spheroids from HGSC ascites, although without relation to the clinical outcomes of the patients [8]. Moreover, there are speculations that ascites spheroids and immune cells in the ascites might reflect some characteristics of the primary tumor and its microenvironment. For primary tumors, it is already confirmed that tumor-infiltrated lymphocytes (TILs) are a favorable prognostic marker for patient survival, whereas tumor-associated macrophages (TAMs) are associated with worse survival outcomes [9]. Programmed death-1 receptor (PD-1), which is predominantly expressed on activated lymphocytes, and its ligand (PD-L1), which is frequently present in tumor cells and some immune cells [10] are important immune checkpoint molecules that may influence the survival of HGSC patients [11]. The presence of PD-1-positive TILs and the expression of PD-L1 in primary ovarian tumors is significantly associated with prolonged overall survival, although the comparison of PD-L1 expression in primary tumors and peritoneal metastatic tissue showed discordant results, and the comparison of PD-1 was not analyzed [12,13,14]. However, the understanding of immune cells and PD-1 and PD-L1 expression in malignant ascites remains completely limited. So far, it is known that immune cells in ascites can be found only in a free-floating state [3]. Notably, there is insufficient information on the presence of immune cells within ascites spheroids of HGSC and the expression of PD-1 and PD-L1 on both tumor and immune cells that constitute these spheroids. Consequently, our objective was to provide a comprehensive description of tumor spheroids in ascites from HGSC patients, focusing on their morphological characteristics, cellular composition, and expression of PD-1 and PD-L1.

## 2. Materials and Methods

### 2.1. Patients

Patients diagnosed with primary HGSC between January 2019 and May 2021 at the Institute of Oncology Ljubljana (IOL) and/or University Medical Centre Ljubljana were included in the study. The inclusion criteria were as follows: age > 18 years, WHO performance status from 0 to 1, histologically confirmed HGSC, International Federation of Gynecology and Obstetrics (FIGO) stage ≥ I A, presence of malignant ascites, and indication for first-line systemic treatment with platinum agents. All patients received standard chemotherapy treatment. Written consent was provided by every patient.

### 2.2. Study Design

Ascites samples were collected at disease presentation, specifically during laparoscopy or laparotomy before tumor biopsy was performed and any treatment was initiated. Ascites samples were immediately sent to the Department of Cytopathology, IOL, where they were processed as previously described by our group [7,15]. The study was designed in order to analyze the size and density of tumor spheroids in HGSC ascites, their composition including tumor cells, immune cells, such as macrophages (CD68+), T lymphocytes (CD3+, together with CD4+ and CD8+ subsets), B lymphocytes (CD20+), capillaries (CD34+ and ERG+ for endothelium, and collagen IV+ for basal membrane), myofibroblasts (alpha smooth muscle actin, αSMA), as well as the expression of PD-1 and PD-L1 on both tumor and immune cells within the spheroids (further referred as spheroid-associated tumor and immune cells). The size and density of spheroids were assessed in Giemsa-stained smears, while the composition of spheroids was assessed from ascites cell block sections by evaluating Hematoxylin and Eosin (H&E) slides and immunocytochemistry (ICC) slides stained for CD3, CD4, CD8, CD20, CD68, CD34, ERG, collagen IV, αSMA, PD-1, and PD-L1. Clinical data were obtained from the patient’s electronic medical record and were used to calculate if there was any correlation between progression-free survival (PFS) and overall survival (OS). For spheroid size and spheroid density, correlation with the FIGO stage and survival of the disease was also calculated. A cut-off value for low and high spheroid density in the ascites was determined to assess better/poorer survival outcomes. Survival analysis of the patients was based on a 3.5-year patient follow-up. A schematic description of the study design is given in Figure 1. The study was conducted in accordance with the Declaration of Helsinki and was approved by the National Ethics Committee in Ljubljana, Slovenia (registration and annex numbers 0120-33/303/2018/3 and 0120-33/303/2018/6, respectively).

### 2.3. Giemsa-Stained Smears and Cell Block Preparation

Ascites samples were centrifuged at 2700 rpm for 10 min to pellet down ascites cells and spheroids. The sediment was primarily used for the preparation of Giemsa-stained smear according to the standard protocol for Giemsa staining at IOL. The residual sediment was fixed and processed to a formalin-fixed paraffin-embedded cell block, as previously described by our group [7,15]. Briefly, 4 μm sections were cut from each cell block for H&E and ICC staining.

### 2.4. Immunocytochemical Staining

ICC staining was performed using following antibodies: CD3 (LN10, 1:500, Leica Biosystems, Wetzlar, Germany), CD4 (SP35, 1:10, Cell Marque, Rocklin, CA, USA), CD8 (144B, 1:100, Agilent, Santa Clara, CA, USA), CD20 (CD20cy, 1:200, Agilent, Santa Clara, California, USA), CD34 (QBEnd, 1:25, Agilent, Santa Clara, CA, USA), CD68 (KP1, 1:2500, Agilent), ERG (EP111, 1:50, Cell Marque, Rocklin, CA, USA), αSMA (1A4, 1:8000, Agilent, Santa Clara, CA, USA), collagen IV (CIV22, 1:20, Cell Marque, Rocklin, CA, USA), PD-1 (NAT-105, 1:200, Cell Marque, Rocklin, CA, USA), and PD-L1 (three different clones: 22C3, 1:50, Agilent, Santa Clara, CA, USA; SP263, RTU, Ventana, Tucson, AZ, USA; SP142, RTU, Ventana, Tucson, AZ, USA). CD3 and CD68 antibodies were combined for dual staining on the same slide section, as well as CD34 and ERG; all other antibodies were used for single staining. The staining was performed on a BenchMark Ultra automated immunostainer (Ventana, Tucson, AZ, USA). Positive reaction for CD3, CD4, CD8, CD20, ERG, collagen IV, αSMA, PD-1, and PD-L1 was evaluated by the presence of diaminobenzidine (DAB) brown precipitate accomplished with OptiVew detection kit (Ventana, Tucson, AZ, USA), while for CD34 and CD68 by the presence of Fast Red–red precipitate accomplished with ultraView Universal Alkaline Phosphatase Red detection kit (Ventana, Tucson, AZ, USA). ICC staining protocols were adopted from the standard immunohistochemical staining protocols at IOL. Negative control slides omitting the primary antibody and appropriate in-house positive control slides for each antibody in the panel were included in all batches. Sections from tonsillar and placenta tissues served as a positive control.

### 2.5. Slide Scanning and Evaluation

Giemsa, H&E, and ICC stained slides were scanned with the NanoZoomer S360MD Hamamatsu C13220 digital slide scanner (Hamamatsu Photonics, Hamamatsu, Japan). Scanned slides were converted to NDPI file format images. Slide images were later analyzed semi-quantitatively/quantitatively using NanoZoomer Digital Pathology Software (Hamamatsu Photonics, Hamamatsu, Japan). One experienced cytopathologist (V.K.P.) evaluated Giemsa-stained slides (V.K.P.), and one experienced pathologist together evaluated H&E and ICC slides (G.G.).

### 2.6. Scoring Criteria

Spheroid size and density were assessed in the Giemsa-stained smears. For the assessment of spheroid size, spheroids were classified into three groups, according to the already published data: small (up to 100 μm), medium (between 100 and 500 μm), and large spheroids (bigger than 500 μm) [16,17]. We estimated spheroid density semi-quantitatively in 10% increments on five randomly chosen power fields on Giemsa-stained smear at 2.5× magnification. Since we found no reports about standardized cut-off values related to spheroid density in ascites or any other effusions in cytology, we conducted a comparison of each 10% increment as a potential cut-off value for spheroid density in relation to survival outcomes of HGSC patients. Moreover, the cell composition of the spheroids was assessed from H&E and ICC slides prepared from cell blocks. Only cell blocks with spheroids present in H&E slides were further analyzed. A correlation between spheroid density in Giemsa-stained smears and spheroid presence in H&E was calculated. Furthermore, a 2.0 mm-dimeter microarray was simulated in ICC slides by randomly choosing a 3.142 mm^2^ area on each sample (in triplicates) to compare equal areas among different-sized cell blocks to achieve the highest reliability with the whole cell block area [18,19,20]. The result was given as an average number (count) of positively stained cells (CD3, CD4, CD8, CD20, CD68, and PD-1) per defined area. Only positively-stained spheroid-associated immune cells were counted, apart from CD4, which was calculated as a subtraction of CD8 from CD3 count due to its positivity on macrophages as well [21]. The presence of capillaries was assessed as present or not present regarding positivity or negativity for CD34 and ERG staining in endothelium and collagen IV for the basal membrane of the capillaries, and the presence of myofibroblasts regarding positivity or negativity of αSMA. PD-L1 was calculated as a percentage of PD-L1 positive spheroid-associated tumor or immune cells per defined area; due to the lack of a standardized cut-off value determining PD-L1 positivity in ovarian cytology, we considered a positive reaction if the PD-L1 score was ≥0.1%. The scoring was performed at 20× magnification on the NanoZoomer Software (Hamamatsu Photonics, Hamamatsu, Japan).

### 2.7. Statistical Analysis

Spearman’s rank correlation coefficient (ρ) was used to describe the correlation between spheroid density in Giemsa-stained smears and spheroid presence in cell block samples, ρ ranged between −1 and +1, where −1 was considered as negative, and +1 as a positive correlation between the analyzed parameters. The median (range) was calculated for the spheroid size, spheroid density, cell count (CD3, CD8, CD4, CD20, CD68, and PD-1), and cell percentage (PD-L1+ cells) within the spheroids. Cronbach α was used to calculate the reliability between PD-L1 clones; α ≥ 0.7 was considered as good reliability. Kaplan–Maier with log-rank test was used to evaluate PFS and OS. PFS was calculated as the time from diagnosis until disease progression or death, and OS was calculated as the time from diagnosis to death. ρ-values and *p*-values < 0.05 were considered significant. Statistical analysis was performed with IBM SPSS Statistic software v28.0.1.0 (142).

## 3. Results

Forty-three patients with primary HGSC were included in the study. The mean age at the time of diagnosis was 61 years (range 41–84 years). According to the International Federation of Gynecology and Obstetrics (FIGO) stage, one patient was classified at stage I, two patients at stage II, 29 patients at stage III, and 11 patients at stage IV.

### 3.1. Evaluation of Spheroid Size and Density

Spheroids were found in 42/43 (97.7%) of Giemsa-stained smears. Interestingly, spheroids in each Giemsa-stained smear were uniformly sized. Therefore, we classified patients’ smears into three main categories based on the estimated size of the spheroids: small (21 patients), medium (12 patients), and large (9 patients). Patients diagnosed at FIGO stages I and II (N = 3) exhibited only small spheroids in the ascites. Among those at FIGO stage III (N = 29), 13 patients had small spheroids, 9 had medium-sized spheroids, 6 patients had large spheroids, and in one patient, only single tumor cells were observed. Meanwhile, at stage IV (N = 11), four patients had small spheroids, three had medium-sized spheroids, and three had large spheroids. Representative images of tumor spheroids with different sizes are shown in Figure 2a–c. Furthermore, the density of the spheroid varied among the patients, and most of the patients (30%) were characterized with ≤ 10 % spheroid density, while the median density was 20 % (range ≤ 10%–100%). The distribution of the spheroid density among the patients is shown in Figure 3b. We tested 10 different increments to see if we could define a cut-off value that might be associated with patient survival. Our results showed that a 10% cut-off value for spheroid density distinguished the patients with significantly different PFS and OS. Hence, patients with ≤ 10% (low) (N = 17, 40%) had significantly better PFS (median 31 vs. 24 months, *p* = 0.039) and OS (median 37 vs. 19 months, *p* = 0.025) compared to patients with >10% (high) spheroid density of tumor spheroids (N = 25, 60%). Kaplan–Meier survival curves are shown in Figure 3c,d. Interestingly, patients diagnosed at the FIGO stage III exhibited a lower median spheroid density (20%) compared to those at FIGO stage IV (40%), but no significant difference among the medians was confirmed (Figure 3a). It is worth mentioning that patients at FIGO stage I (N = 1) and patients at FIGO stage II (N = 2) were characterized by a median spheroid density of ≤ 10%, but due to the small number of patients, we could not significantly confirm a relevant median of spheroid density in these groups.

The presence of tumor spheroids in cell blocks from ascites was identified in only 25/43 (58%) patients. Actually, we observed a significant correlation between spheroid density in Giemsa-stained smears and spheroid-containing cell blocks, meaning that only high-density tumor spheroids in Giemsa-stained smears were further present in the corresponding cell blocks (ρ = 0.618, 95% CI = 0.382–0.778, *p* < 0.001). Patients with low spheroids density were most likely characterized by single, dissociated tumor cells in the cell blocks, and those were excluded for further analysis. Interestingly, patients with medium- and large-sized spheroids were also most likely to be detected in the cell blocks (ρ = 0.043), which was not the case with small-sized spheroids.

### 3.2. Evaluation of Spheroid Cell Composition

Only 25 (58%) patients with tumor spheroids present in the cell blocks were further evaluated for assessing spheroid-associated immune cells and capillaries. Median (range) values for spheroid-associated immune cells, capillaries, and myofibroblasts are given in Table 1. We observed spheroid-associated T lymphocytes in 20/25 of analyzed patients (median cell count of 20 cells per area (range 0–119)). Most of them were CD8 subtypes. Spheroid-associated B lymphocytes were present only in 5/25 patients. The amount of spheroid-associated B lymphocytes was very low (median cell count of 0 cells per area (range 0–7)). In fact, most of the CD20+ positively stained cells in the cell blocks were not part of the tumor spheroids but were free floating (20/25 patients). Spheroid-associated macrophages were observed in 11/25 patients and were the second most dominant immune population within the spheroids (median cell count two cells per area (range 0–113)). PD-1 expression was observed in 8/25 patients. However, the amount of PD-1+ immune cells was very low, with a median close to 0 cells per area (range 0–49). Noteworthy, only lymphocytes were positive for PD-1. Interestingly, PD-1 was expressed only by spheroid-associated lymphocytes, which was not the case with the remaining free-floating lymphocytes. PD-1 expression on macrophages and tumor cells was not observed. The presence of capillaries (both endothelium and basal membrane) was detected in three out of 25 patients. Interestingly, we noticed ERG-positive spheroid-associated tumor cells, which is an unexpected finding. To add, we did not observe the presence of myofibroblasts within the spheroids of the analyzed patients. However, no correlation between any of the spheroid-associated immune cell counts and patient outcome was observed, nor was the presence of capillaries and ERG+ spheroid-associated tumor cells. Representative images of H&E and ICC stains for spheroid-associated immune cells and PD-1 expression are shown in Figure 4, and the presence of capillaries (endothelium and basal membrane) in HGSC ascites are shown in Figure 5.

### 3.3. Expression of PD-L1 on Spheroid-Associated Immune and Tumor Cells

The expression of PD-L1 on spheroid-associated immune and tumor cells was assessed by comparing the staining results of three different antibody clones: 22C3, SP263, and SP142 (Figure 6). PD-L1 expression was observed on spheroid-associated immune cells in 19/25 patients with the 22C3 clone and 14/25 patients with the SP263 clone, whereas on tumor cells in 11/25 and 10/25 patients, respectively; staining with the SP142 clone was negative for both immune and tumor cells. In general, the expression of PD-L1 on immune and tumor cells was very low, with a median expression of ≤0.1% per area (range 0–10%). Staining with 22C3 and SP263 clones showed good reliability when comparing staining results on both immune (α = 0.80) and tumor cells (α = 0.77). However, the SP263 clone stained slightly more immune cells (1.2 ± 1.9 %) and fewer tumor cells (0.5 ± 1.3 %) compared to the 22C3 clone. A detailed description of the PD-1 and PD-L1 expression per patient is given in Table 2. Since the number of cases included in PD-1 and PD-L1 analysis was very low, to calculate their correlation with patient outcome, we grouped the results in no expression and any expression present (≥0.1%). The analysis showed no significant correlation between PD-L1 expression on spheroid-associated tumors and immune cells with patient outcomes. However, we observed a trend towards better OS in patients with present PD-L1 expression on ICs, assessed by 22C3 and SP236 clones (Figure 7a,b).

## 4. Discussion

HGSC spheroids are considered bona fide metastatic units that attach to the mesothelium or free float in malignant ascites and are responsible for further invasion and dissemination [22]. Complex in vitro and ex vivo 3D models have been used for spheroid research and mostly for developing novel tools to test different drugs and therapies [23]. Consequently, it is still not clear what are their cellular characteristics and clinical significance in the ascites itself. Here, we analyzed spheroids in Giemsa-stained smears and cell blocks of HGSC ascites and outlined their morphological features, cell composition, including capillaries, together with PD-1 and PD-L1 expression on spheroid-associated tumor and immune cells, as well as their impact on the patient’s survival.

It is generally accepted that spheroids in ascites differ in size and shape. However, there is no consensus on how to categorize spheroids based on their size in the ascites of clinical patient samples, and consequently, the impact of the size on the patient’s survival remains unclear. We found one study to describe the distribution of the spheroid size in HGSC ascites, where the majority of the spheroids were smaller, between 50 and 75 µm, and also larger spheroids up to 1 mL were present (median 55 µm) [8]. However, this study did not specify a concrete cut-off for the classification of spheroid size. We found size criteria for spheroids exclusively in studies focusing on spheroid 3D models [16,17]. Thakuri et al. have reported a quantitative size-based analysis of in vitro spheroids for drug testing where they defined three main categories of spheroid sizes: small (up to 100 μm), medium (between 100 and 500 μm), and large (bigger than 500 μm) [16]. We applied this size criteria for our analysis and confirmed the presence of all three spheroid sizes. Interestingly, we noticed that the size of the spheroids in each patient’s Giemsa-stained smear appeared to be uniformly consistent, and patients characterized with small spheroids in the ascites were more common than patients with medium or large spheroids. Small spheroids were the most common in our study in 45% of the analyzed patients. This result was similar to the study of Micek et al. that confirmed the predominance of small spheroids in their patient cohort. However, we did not find any correlation between the spheroid size and the patient’s survival outcomes or with the FIGO stage. It is important to note that the three patients in FIGO stages I and II exhibited only small spheroid sizes. Patients in these stages are rarely diagnosed with ascites, making it challenging to gather enough cases within the same patient cohort for comparison with FIGO stages III and IV. For patients in FIGO stages III and IV, we did not observe significant differences, although the average size of spheroids in FIGO stage III patients appeared to be slightly lower than that in FIGO stage IV patients. Top of Form.

On the other hand, it was already confirmed that a larger ascites volume significantly correlates with a worse prognosis of HGSC patients [24], but no reports show how the density of spheroids in ascites is related to disease progression. For this reason, we investigated whether we could establish a cut-off value for spheroid density in ascites that could predict patients at higher risk for better or worse survival. We tested different cut-off values in 10% increments and showed that a cut-off value of 10% for spheroid density in Giemsa-stained smears significantly stratified patients into two groups: patients with spheroid density above the established cut-off value were associated with worse PFS and OS, whereas patients below the cut-off value showed the opposite survival pattern. To add, the three patients in FIGO stage I and II were associated with a low spheroid density of below 10%, while the patients in FIGO stage IV exhibited insignificantly higher median spheroid density (40%) compared to stage III (20%). This finding suggests that the use of routinely obtained Giemsa-stained smears could provide valuable prognostic information related to patient outcomes, although a larger patient cohort is required to investigate it further, as well as the findings related to the spheroid size.

In the other part of the study, our aim was to investigate spheroids and their cell composition in cell blocks. Cell blocks have an advantage over Giemsa-stained smears as they allow examination of all depth levels within the 3D structure of spheroids, enabling a detailed examination of the cellular composition. Our results showed that spheroids were present in cell blocks only in cases with large spheroids and high spheroid density. Otherwise, dissociated tumor cells were found instead of spheroids. We speculate that the preparation process of the cell blocks, involving multiple centrifugations, fixation, and pipetting steps, may have disrupted the spheroid structure in the cell blocks. Consequently, we confirmed the presence of spheroids in cell blocks in 58% of all patients in our cohort. Apart from our study, we found only three other studies that investigated ascites spheroids of HGSC in cell blocks [8,25,26]. Capellero et al. used immunofluorescence of PAX8, EpCAM, and Ki-67 to confirm the identity of tumor cells that formed the spheroids in the ascites of HGSC patients [25]. In a previous study conducted by our group, we tested the same and other HGSC markers on cell block sections obtained from these patients [7]. Using immunocytochemistry, we compared the morphological and immunocytochemical features of tumor cells in ascites with those of primary tumors. Considering the positive results and experience from this previous study, we decided to use immunocytochemistry for this analysis as well.

Until now, it was generally accepted that only tumor-associated macrophages are located in the center of spheroids surrounded by tumor cells, and T lymphocytes are present in a free-floating state [3]. T lymphocytes are of great interest for research since TILs in primary solid HGSC tumors are being associated with a better patient prognosis. Here, we confirmed the presence of lymphocytes in ascites spheroids of HGSC. Apart from our findings, we found only one other study by Iwahashi et al. to deny the misinformation that T lymphocytes are only present as free-floating cells in the ascites [26]. In fact, we observed the presence of both T and B lymphocytes in spheroids. T lymphocytes were the most abundant of all spheroid-associated immune cells, with CD8+ dominating over the CD4+ subset, while B lymphocytes were present mainly in a free-floating state outside of the spheroids. In another previous study of our group, where we examined all immune cells by flow cytometry in ascites samples from the same patient cohort used here, we noticed a clear dominance of CD4+ over CD8+ subsets, indicating that CD8+ T lymphocytes are more likely to be localized within the spheroids than being present in a free-floating state [27]. Additionally, B lymphocytes accounted for less than 2% of the total leukocyte population in the ascites fluid, indicating a generally low presence of B lymphocytes in HGSC ascites [27]. Otherwise, Iwanishi et al. focused only on the CD8+ subset and reported a correlation between CD8+ spheroid-associated lymphocytes (assessed immunocytochemically) and patient prognosis [26]. However, our findings were contrary to theirs, as we did not observe a significant association between any of the spheroid-associated lymphocytes and patient survival. Noteworthy, their study included 10 patients, while ours had 25—in both cases, the number of patients was relatively low, which might affect the statistical significance of both analyses.

Furthermore, our results confirmed the presence of spheroid-associated macrophages in cell blocks, as did Capellero in their study, where they additionally looked in spheroid-associated macrophage subsets, observing a higher abundance of M2 pro-tumorigenic type over M1 pro-inflammatory type [25]. Contrary to these reports in cell blocks, in our previous flow-cytometry study, we revealed a higher presence of M1-like macrophages compared to M2-like macrophages in a free-floating state in ascites [27]. It is worth mentioning that these results were discordant with the general published data on M2-like macrophage predominance in HGSC ascites. We explain this discrepancy by differences in inclusion criteria in published studies. In most published studies, patients with advanced stage of the disease or recurrent disease are included, while in our study, we included only patients at diagnosis of HGSC—prior to initiation of any oncological treatment [27]. Nevertheless, a study by Long et al. showed that tumor cells and macrophages have stronger interactions within the spheroid compared to the dissociated state [28], which might explain the correlation between both findings mentioned above. Furthermore, Long et al. also demonstrated that when HGSC tumor cells were part of the spheroid, they promoted the polarization of macrophages toward the M2-type phenotype. A drawback of our study was the lack of a detailed analysis of the polarization type of spheroid-associated macrophages in HSGC ascites.

We also confirmed PD-1 positive ICs within the ascites spheroids. In fact, PD-1 was mainly expressed in spheroid-associated lymphocytes and not in free-floating lymphocytes. This was apparent from the morphology of the PD-1+ stained cells. They appeared to be more similar to lymphocytes, as they were smaller than tumor cells and macrophages and, unlike macrophages, lacked dendritic extensions. De la Fuente et al. investigated the expression of PD-1 in advanced HGSC primary tumors and reported that it was almost exclusively expressed by lymphocytes, which is consistent with our findings in the ascites [20]. In their study, the expression of PD-1 was low in 70% (31/130) of the patients, and it was correlated with better OS. However, we did not observe any correlation between lymphocyte count and PD-1 expression with the patient outcome. Since the number of cases included in our analysis was very low, that could probably be the reason why we did not confirm any impact of spheroid-infiltrated lymphocytes on patient survival either. Regarding PD-L1 expression, there are not completely clear data on PD-L1 expression, evaluation criteria, and its clinical relevance in primary ovarian carcinoma. In the IMAGYN050 trial, it was shown that almost 2/3 of newly diagnosed stage III or IV ovarian carcinomas had a moderate PD-L1 expression (assessed by SP142 clone on immune cells), which was associated with the worst prognosis, mainly on immune cells rather than tumor cells. Less than 25% of the patients in the trial demonstrated >5% PD-L1+ immune cells [29]. De la Fuente et al. reported PD-L1 expression in advanced HGSC by FDA-approved 22C3 clone mainly in macrophages (not in tumor cells) and improved OS in the patient group with high PD-L1 [20]. In our case, in the ascites, low PD-L1 expression was seen on both tumor cells, lymphocytes, and macrophages assessed by antibodies 22C3 and SP263 clones, which correlate with the PDL-1 expression reported in primary HGSC. We found the study of Iwahashi et al. to be the only one to look into PD-L1 expression on CD8 T lymphocytes in ascites spheroids, except for our study [26]. Similar to us, they evaluated PD-L1 expression in ascites cell blocks in 58% of their patient cohort and reported them to be equivalent to corresponding tumor tissue, but interestingly, they did not perform a prognostic analysis of PD-L1. Since they used different assessment criteria to judge the immunochemical analysis, it was difficult to compare the results of both studies. In our study, we did not identify any correlation with patient survival, most likely due to the small number of cases included.

Remarkably, we have verified the presence of capillaries (comprising of endothelium and basal membrane) within the ascites spheroids in the same three patients. To our knowledge, no similar publications were found on this subject. Capillaries were identified as double ERG and CD34-positive endothelial cells surrounded by collagen IV-positive basal membrane. Unexpectedly, we also observed the presence of ERG-positive and CD34-negative spheroid-associated tumor cells. ERG expression was described in prostate tumors but was not reported to be associated with ovarian carcinomas [30,31]. The angiogenesis itself could potentially lead to a more aggressive disease progression, resulting in the development of bulky abdominal conditions; however, we did not find any correlation with patient outcomes. We also did not confirm the presence of myofibroblasts within the spheroids, which is consistent with previously published data [8]. We believe that a further study including a higher number of patients is essential to re-evaluate the real impact of the capillaries on patient survival, and of course, an additional investigation is required to explain the background of ERG expression in tumor cells of HGSC.

## 5. Conclusions

In our study, we observed the existence of spheroids in different sizes and densities in ascites of HGSC patients. We established a significant cut-off value for spheroid density to predict patient survival, showing better survival outcomes associated with patients characterized with less than 10% of spheroid density. Moreover, we confirmed the presence of spheroid-associated immune cells as well as the expression of PD-1 on spheroid-associated lymphocytes and PD-L1 expression on both spheroid-associated immune and tumor cells. Also, we observed the presence of capillaries within the spheroids. These findings highlight the promising potential of routine cytopathological techniques of HGSC spheroids in ascites as a valuable tool for investigating them as potential prognostic markers.

## Figures and Tables

**Figure 1 cells-12-02390-f001:**
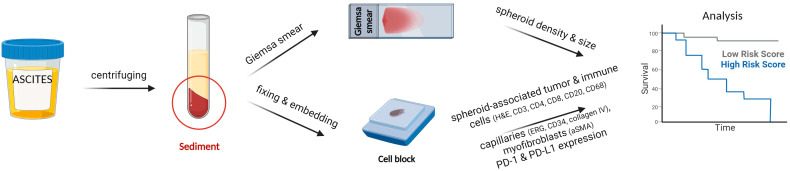
Study design scheme. Ascites samples were centrifuged to pellet down ascites cells and spheroids. The sediment was primarily used for Giemsa-stained smear preparation, while the residual sediment was further processed to a cell block. Giemsa-stained smears were used to describe spheroid density and size, and cell blocks were used to evaluate spheroid-associated tumor and immune cells and capillaries, as well as PD-1 and PD-L1 expression. Obtained data were used for the survival analysis. Created with BioRender.com.

**Figure 2 cells-12-02390-f002:**
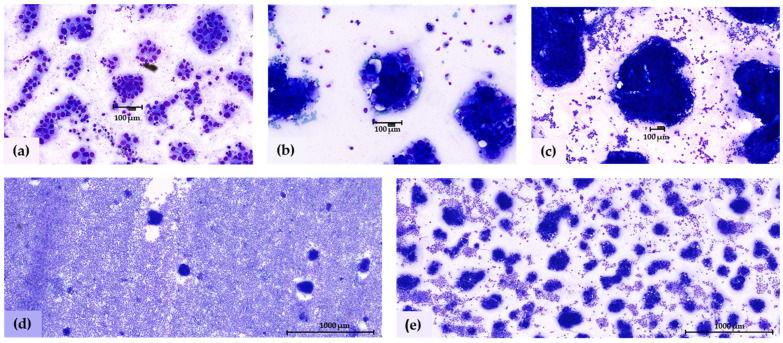
(**a**–**c**) Spheroid size and density, assessed in ascites Giemsa-stained smears from HGSC patients. Representative images of (**a**) small (100× magnification), (**b**) medium (100× magnification), and (**c**) large spheroids (50× magnification). The black ruler represents 100 μm length. (**d**,**e**) Two different spheroid density groups of spheroids were detected in Giemsa-stained smears. Representative images of (**d**) less than 10% of spheroid density and (**e**) 80% of spheroid density. The black ruler represents 1000 μm length.

**Figure 3 cells-12-02390-f003:**
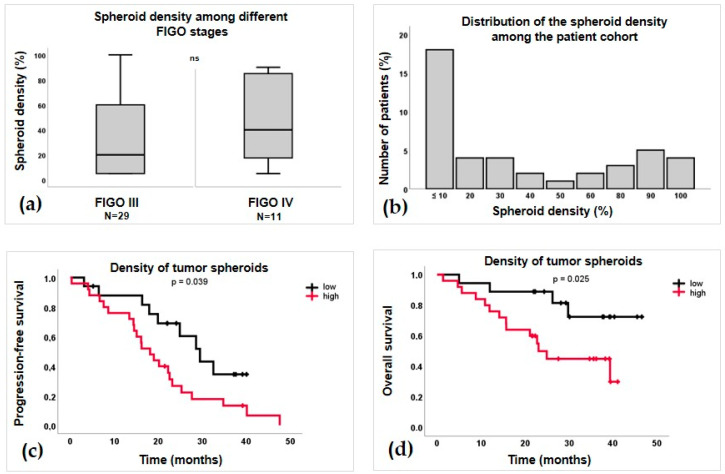
(**a**) Box plots showing the distribution of spheroid density among patients diagnosed at FIGO stages III and IV. (**b**) Distribution of the spheroid density among the whole patient cohort. (**c**,**d**) Correlation of the density of tumor spheroids with patient survival. (**c**) Progression-free survival. (**d**) Overall survival.

**Figure 4 cells-12-02390-f004:**
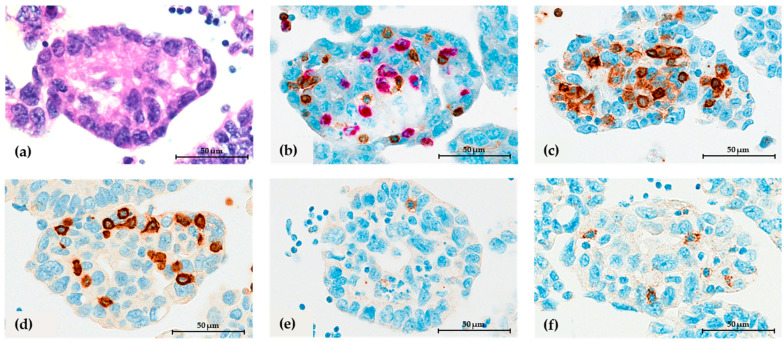
Spheroid-associated immune cells in HGSC ascites. (**a**) H&E staining. (**b**) Dual immunocytochemical (ICC) staining for T lymphocytes (CD3+) in brown and macrophages (CD68+) in red. Single ICC for (**c**) CD4+ T subsets, (**d**) CD8+ T subsets, (**e**) CD20+ lymphocytes B, and (**f**) PD-1 in lymphocytes. The black ruler represents 50 μm length.

**Figure 5 cells-12-02390-f005:**
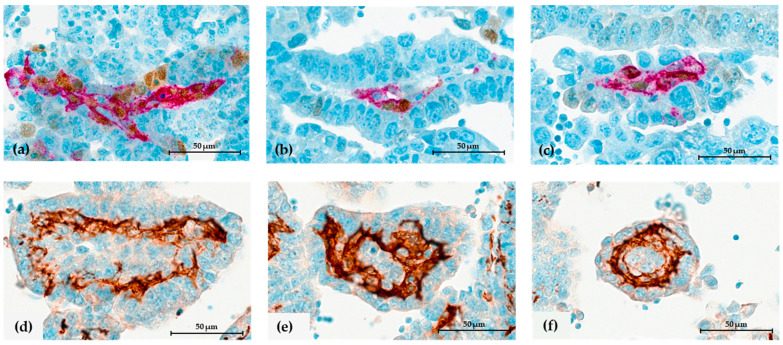
Capillaries within the spheroids in HGSC ascites. (**a**–**c**) Dual immunocytochemical (ICC) staining for ERG (brown) and CD34 (red), indicating the presence of capillary endothelium (ERG+CD34+) and ERG+ tumor cells within the spheroids. (**d**–**f**) Single ICC for collagen IV (collagen IV+), indicating capillary basal membrane (400× magnification). The black ruler represents 50 μm length.

**Figure 6 cells-12-02390-f006:**
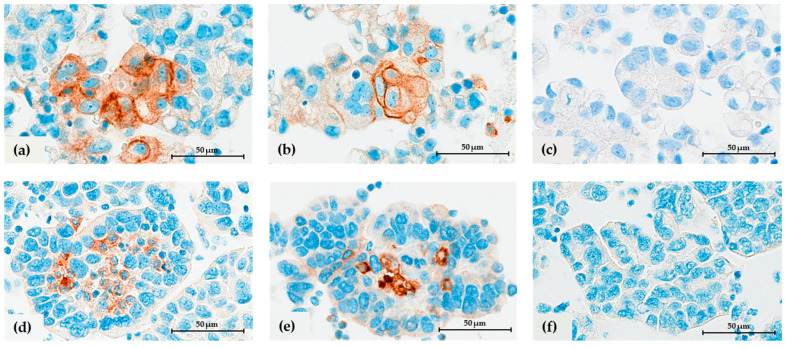
Expression of PD-L1 on spheroid-associated tumor cells (**a**–**c**) and immune cells (**d**–**f**) by immunostaining with 22C3 (**a**,**d**), SP263 (**b**,**e**), and SP142 (**c**,**f**) clones at 400× magnification**.** The black ruler represents 50 μm length.

**Figure 7 cells-12-02390-f007:**
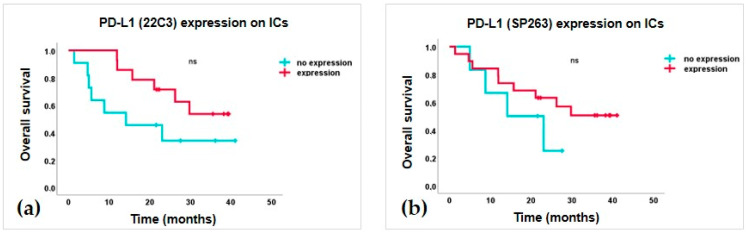
Survival curves represent a trend towards better overall survival (OS) for patients with higher expression of PD-L1 on spheroid-associated immune cells, assessed by (**a**) 22C3 and (**b**) SP263 antibody clones, respectively.

**Table 1 cells-12-02390-t001:** Median (range) count values for spheroid-associated immune cells and capillaries per defined cell block area (N = 25).

Cell Type	T Lymphocytes	T Lymphocyte Subsets		B Lymphocytes	Macrophages	PD-1+ Immune Cells	Capillaries		Myofibroblasts
ICC marker	CD3	CD8	CD4	CD20	CD68	PD-1	CD34, ERG (endothelium)	collagen IV (basal membrane)	αSMA
Positive cases N, (%)	20 (80)	18 (72)	11 (44)	5 (20)	11 (44)	12 (48)	3 (12)	3 (12)	0 (0)/0 (0)
Median cell count (range)	7 (0–119)	4 (0–118)	1 (0–26)	0 (0–7)	2 (0–130)	0 (0–42)	NA	NA	NA

Note: the cell count of the immune cells and capillaries is given per area (3.142 mm^2^ of each cell block slide). Abbreviations: ICC, immunocytochemistry; NA, not applicable.

**Table 2 cells-12-02390-t002:** Median (range) percentage for PD-L1 expression on spheroid-associated immune and tumor cells, assessed in the cell blocks (N = 25).

Cell Type	Spheroid-Associated Immune Cells			Spheroid-Associated Tumor Cells		
PD-L1 clone	22C3	SP263	SP142	22C3	SP263	SP142
N positive cases, N (%)	14 (56)	19 (76)	0 (0)	10 (40)	11 (44)	0 (0)
Median PD-L1 % (range)	0.1 (0–10)	1 (0–10)	0 (0)	0 (0–10)	0 (0–5)	0 (0)

Note: the percentage of spheroid-associated immune and tumor cells is given per area (3.142 mm^2^ of each cell block slide).

## Data Availability

All relevant data regarding this manuscript are available from the above-listed authors.

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
