# Peer review of "Morphological and Immunocytochemical Characterization of Tumor Spheroids in Ascites from High-Grade Serous Carcinoma"

_cells, 2023, doi:10.3390/cells12192390_

Round 1

Reviewer 1 Report

The study by Miceska et al. has a well-defined objective of characterizing HGSC ascites across several parameters and examining correlations  with patient survival. The study design is clear and effective for these objectives. A strength of the data set is that the samples are limited to HGSC patients who have not undergone any treatment, thereby reducing heterogeneity to some extent.

The paper would be strengthened by the addition of some clarifying details and more refined analysis of the data.

1)    The authors state that there is no qualitative/quantitative study of spheroids and their clinical significance. This is true, although some data on spheroids without relation to clinical outcome was recently published (Micek 2023, PMID: 36875739). That paper provided the size of spheroids, percentage of cells in spheroids, and cell types/ECM found in spheroids from a smaller clinical cohort. 

2)    Please provide a clear definition for "spheroid density" and how it was measured.

3)    Clarify any relationships between the data and disease stage

a.     For example, were most of the patients with small spheroids from the early stages and large spheroids from stage IV? This might explain the difference in size compared to the Micek cohort (where spheroids appeared to be smaller in general, but most patients were stage III).

b.     Correlating the spheroid density with the patients' staging would provide context to the result of low-density tumor spheroids correlating with better PFS and OS than high-density tumor spheroids. In particular, does this relationship hold if you remove the stage I/II cases which should have a much better PFS/OS.

4)     The presence of capillaries in the spheroids is fascinating. However, from the representative image Fig 3f, the size of the spheroid looks significantly greater, resembling more like a tissue fragment. Tumor spheroids within ascites typically have either round or elliptical shapes with some curvature, as shown in Fig. 2 and 3(a-e) within the paper. If other cases have CD34+ERG+ staining within a distinct tumor spheroid, that will add more weight to the claim of potential angiogenesis within HGSC spheroids.

5)     Please use more precise terminology to describe how often spheroids have other cell types. For example, when you say 20/25 cases had T cells, is that 20 unique patients or 20 spheroids? If 20 patients, how common were T-cell positive spheroids within a patient?  Is cell count the #/spheroid or #/section (which may have multiple spheroids)? In general, presenting the data in more nuance (histogram vs. just median) would be useful. Also, is there a correlation in the count of T lymphocytes (CD3+) per spheroid with the patient stages/outcomes? Similar correlations with staging can be made for spheroid-associated macrophages as well. 

6)     The paper claims that PD-1 is only expressed by spheroid-associated lymphocytes, with no expression being observed on macrophages and tumor cells (lines 229-231). Are there any co-stained images to demonstrate this cell type specificity more clearly?

7)     Some minor errors in figures and text:

o   Add scale bars to Figures 3 and 4

o   In the legend for Figure 3 there is a reference to a panel g? should this be f?

o   Typing error on line 157 - The reference [118-20] is perhaps [18-20]

o   Typing error on line 365 - "draft-back" is perhaps "drawback"o   Typing error on line 235 - "PD-L1 expression" should be "PD-1 expression"?

Author Response

1)    The authors state that there is no qualitative/quantitative study of spheroids and their clinical significance. This is true, although some data on spheroids without relation to the clinical outcome was recently published (Micek 2023, PMID: 36875739). That paper provided the size of spheroids, percentage of cells in spheroids, and cell types/ECM found in spheroids from a smaller clinical cohort. 

Thank you for sharing this published study with us. We have reviewed the paper and have included it in our manuscript.

2)    Please provide a clear definition for "spheroid density" and how it was measured.

Thank you for your comment. We have included a more precise definition of spheroid density, specifying that it was assessed semi-quantitatively in 10% increments based on observations from five randomly selected fields on Giemsa-stained smears at 2.5x magnification. Since we couldn't find any reports providing standardized cut-off values for spheroid density in ascites or other cytological effusions, we proceeded to compare each 10% increment as a potential cut-off value for spheroid density in relation to the survival outcomes of HGSC patients.

3)    Clarify any relationships between the data and disease stage.

  1. For example, were most of the patients with small spheroids from the early stages and large spheroids from stage IV? This might explain the difference in size compared to the Micek cohort (where spheroids appeared to be smaller in general, but most patients were stage III).

Thank you for this suggestion. We have provided a detailed description of the distribution of spheroid size among all patients in our cohort and compared the data among different FIGO stages, revealing that small spheroids were the most common, consistent with the findings of Micek in 2023. In fact, among the patients diagnosed at FIGO stages I and II (N=3), only small spheroids were observed in the ascites. Among those at FIGO stage III (N=29), 13 patients had small spheroids, 9 had medium-sized spheroids, 6 had large spheroids, and in one patient, only single tumor cells were observed. Meanwhile, at stage IV (N=11), four patients had small spheroids, three had medium-sized spheroids, and three had large spheroids. However, due to the limited number of patients in each FIGO category (especially in stages I and II) we were unable to confirm any significant differences.

  1. Correlating the spheroid density with the patients' staging would provide context to the result of low-density tumor spheroids correlating with better PFS and OS than high-density tumor spheroids. In particular, does this relationship hold if you remove the stage I/II cases which should have a much better PFS/OS.

Patients, diagnosed at the FIGO stage III exhibited a lower median spheroid density (20%) compared to those at FIGO stage IV (40%) but no significant difference among the medians was confirmed (Figure 3, a). It is worth mentioning that the patients at FIGO stage I (N=1) and patients at FIGO stage II (N=2) were characterized by a median spheroid density of ≤ 10% but due to the small number of patients we could not significantly confirm a relevant median of spheroid density in these groups. The inclusion/exclusion of the patients at FIGO I and II did not affect the PFS/OS status of the whole patient cohort, again due to the very low number of patients in this category (N=3).

4)     The presence of capillaries in the spheroids is fascinating. However, from the representative image Fig 3f, the size of the spheroid looks significantly greater, resembling more like a tissue fragment. Tumor spheroids within ascites typically have either round or elliptical shapes with some curvature, as shown in Fig. 2 and 3(a-e) within the paper. If other cases have CD34+ERG+ staining within a distinct tumor spheroid, that will add more weight to the claim of potential angiogenesis within HGSC spheroids.

Thank you for your observation. The selection of spheroids representing capillaries was done randomly. To enhance clarity, we have included three representative images in this regard. Additionally, in response to the request of the other reviewer, we conducted additional ICC staining for collagen IV, which further confirmed the presence of capillary basal membranes alongside capillary endothelial cells (ERG+CD34+). It's worth noting that we also performed staining for myofibroblasts using an α-SMA antibody, but we did not confirm the presence of myofibroblasts in the spheroids.

5)     Please use more precise terminology to describe how often spheroids have other cell types. For example, when you say 20/25 cases had T cells, is that 20 unique patients or 20 spheroids? If 20 patients, how common were T-cell positive spheroids within a patient?  Is cell count the #/spheroid or #/section (which may have multiple spheroids)? In general, presenting the data in more nuance (histogram vs. just median) would be useful. Also, is there a correlation in the count of T lymphocytes (CD3+) per spheroid with the patient stages/outcomes? Similar correlations with staging can be made for spheroid-associated macrophages as well. 

We have addressed this lack of clarity by replacing "cases" with "patients" and the number of cells per defined area (detailly explained in the method section) and by including histograms that provide more precise information regarding our results. Regarding your inquiry on the correlation of each cell count, we would like to mention again that the number of patients in each FIGO stage was very low for giving statistically relevant information in this regard.

6)     The paper claims that PD-1 is only expressed by spheroid-associated lymphocytes, with no expression being observed on macrophages and tumor cells (lines 229-231). Are there any co-stained images to demonstrate this cell type specificity more clearly?

The expression of PD-1 on lymphocytes was visually evident from the morphology of the PD-1+ stained cells. These cells appeared more similar to lymphocytes, as they were smaller in size compared to tumor cells and macrophages, and unlike macrophages, they lacked dendritic extensions. Based on the assessment of our experienced cytopathologist, this was deemed sufficient for drawing a conclusion about the cell type.

7)     Some minor errors in figures and text:

Add scale bars to Figures 3 and 4

  • In the legend for Figure 3 there is a reference to a panel g? should this be f? - corrected
  • Typing error on line 157 - The reference [118-20] is perhaps [18-20] corrected
  • Typing error on line 365 - "draft-back" is perhaps a "drawback" Typing error on line 235 - "PD-L1 expression" should be "PD-1 expression"? - corrected

Reviewer 2 Report

The present article, written by Miceska and colleagues, provides an interesting characterization of OC ascites with important correlations between OC spheroids density, PD-1/PD-L1, and OS in patients.

It could have been of interest a more insightful characterization of other populations of cells and extracellular matrix components present in spheroids (if any) highly involved in spheroids compaction, such as cancer associated fibroblasts, fibronectin, collagen so on, which are correlated with tumorigenicity and poor OS.

In the text, the use of high-grade serous ovarian carcinoma (HGSOC) instead of HGSC better defines the studied model.

Author Response

The present article, written by Miceska and colleagues, provides an interesting characterization of OC ascites with important correlations between OC spheroids density, PD-1/PD-L1, and OS in patients.

1) It could have been of interest a more insightful characterization of other populations of cells and extracellular matrix components present in spheroids (if any) highly involved in spheroids compaction, such as cancer-associated fibroblasts, fibronectin, collagen so on, which are correlated with tumorigenicity and poor OS.

Thank you for this intriguing suggestion. In response, we conducted additional ICC staining for collagen IV and αSMA. We confirmed the presence of the basal membrane of the capillaries within the spheroids based on the positivity of collagen IV. However, as we observed no αSMA expression whatsoever, we have ruled out the presence of myofibroblasts. Unfortunately, we did not have access to other antibodies to conduct further analysis of different ECM components. Nevertheless, we believe that our supplementary work adequately addresses this inquiry.

2) In the text, the use of high-grade serous ovarian carcinoma (HGSOC) instead of HGSC better defines the studied model.

We followed the nomenclature guidelines provided by the WHO, and we have included a scan of the latest WHO Classification of Tumours Book as supporting evidence. Nevertheless, if there is a concern or suggestion regarding the name and its abbreviation, we are open to considering a change.

Reviewer 3 Report

This article is related to one of the most cutting edge problem in oncogynecoloogy, high grade serous carcinomas. It is well known that ascites plays the significant role in this disease progression and relapse and some authors even suppose that it can contain the tumor niches to support high grade serous carcinoma cells. Nevertheless, the true role of tumor spheroids has yet to be investigated and this article impactfully contributes into the spheroids pathogenetic role and their physiology, including angiogenesis. Although the  cell blocks usage for PD-L1 expression assessment is controversial from my point of you, because we could not evaluate the proportion of tumour cells, lymphocytes, macrophages, and other cells in a cell block properly to extrapolate these data to spheroids and even more to metastases.

Author Response

there was no reviewer 3

Round 2

Reviewer 1 Report

The authors have addressed all comments and the expanded analysis improves the utility of this study for others interested in this topic. 

Reviewer 2 Report

Minor points addressed.

Thank you. Congratulations!